# A Qualitative Study Exploring the Rehabilitation Experience of Individuals with a Previous Diagnosis of Cancer and/or Sepsis, Their Caregivers, and Health Providers

**DOI:** 10.3390/healthcare13070822

**Published:** 2025-04-04

**Authors:** Jenna Smith-Turchyn, Christopher Farley, Anastasia N. L. Newman, Jayden Pannu, Bram Rochwerg, Som D. Mukherjee, Marla Beauchamp, Linda C. Li, Hira Mian, Michelle E. Kho

**Affiliations:** 1School of Rehabilitation Science, McMaster University, Hamilton, ON L8S 1C7, Canada; farlec3@mcmaster.ca (C.F.); newmanan@mcmaster.ca (A.N.L.N.); pannuj9@mcmaster.ca (J.P.); beaucm1@mcmaster.ca (M.B.); khome@mcmaster.ca (M.E.K.); 2Department of Medicine, McMaster University, Hamilton, ON L8P 1H6, Canada; rochwerg@mcmaster.ca (B.R.); hira.mian@medportal.ca (H.M.); 3Department of Oncology, McMaster University, Hamilton, ON L8V 5C2, Canada; mukherjee@hhsc.ca; 4Department of Physical Therapy, University of British Columbia, Vancouver, BC V6T 1Z3, Canada; lli@arthritisresearch.ca

**Keywords:** chronic illness experience, health promotion, cancer, sepsis, rehabilitation, patient education

## Abstract

Background/Objectives: Survivors of cancer have more than double the risk of developing sepsis compared to those with no history of cancer. Those who develop sepsis have lasting side effects reducing their physical function and quality of life. Rehabilitation-related needs and barriers are unknown for individuals with cancer who have had sepsis. The aim of this study was to examine the rehabilitation-related experiences of patients with cancer and/or sepsis, their caregivers, and the healthcare team and the educational needs regarding the rehabilitation of patients with sepsis. Methods: We performed a qualitative descriptive study using interviews and focus groups for data generation. We included adults living in Canada who were English-speaking, currently or formerly diagnosed with cancer and/or sepsis, or a caregiver of someone with a current or past diagnosis of cancer and/or sepsis, or a healthcare professional working with this population. Two reviewers used NVivo software for data management and conducted a qualitative data analysis. Results: We included 30 study participants (nine patients, nine caregivers, 12 healthcare professionals; 15 for one-on-one interviews; 15 in the focus groups). We identified three overarching themes relating to rehabilitation: (1) the rehabilitation experience did not meet the patient’s care expectations; (2) barriers to rehabilitation exist on multiple levels; (3) there are important gaps in education on how to improve physical and psychosocial outcomes. We identified two themes related to educational resources: (1) the educational content provided must be specific and meaningful; (2) multi-modal resources are needed to suit diverse partner needs. Conclusions: We identified rehabilitation barriers related to healthcare settings, the pandemic, and workplace culture for those with cancer and sepsis. This study highlights the need to create diverse educational resources on rehabilitation for those with sepsis to improve outcomes and patient/caregiver satisfaction.

## 1. Introduction

Sepsis is broadly defined as a life-threatening illness caused by a dysregulated immune response to infection [1,2]. Each year, sepsis affects approximately 49 million people worldwide, with approximately 38 million individuals surviving hospitalization for sepsis each year [3]. However, many of these individuals are re-hospitalized or die within 90 days of discharge, posing a significant healthcare problem [4]. The lack of adequate rehabilitation exacerbates this issue.

More than 20% of sepsis hospitalizations are cancer-related [1,2]. Cancer is a common condition in adults; two in five Canadian adults will be diagnosed with cancer [5]. Survivors of cancer have more than double the risk of developing sepsis over their lifetime compared to those with no history of cancer [1,2]. This increased risk of sepsis in those with a history of cancer is often multifactorial and may be related to cancer treatments (such as chemotherapy, targeted therapies, radiation and/or steroids, which can all suppress the immune system) and procedures (e.g., surgery, central venous access insertion, biopsies) used to treat and diagnose cancer [1,2]. Sepsis is associated with 5% of all cancer deaths [2].

Approximately 50% of those surviving sepsis will develop post-sepsis syndrome, defined as long-term physical, psychological, or cognitive medical issues [6,7,8,9,10]. Common symptoms persisting for long periods of time after hospital discharge include fatigue, shortness of breath, muscle and joint pain, difficulty sleeping, poor appetite, depression, anxiety, difficulty concentrating, and post-traumatic stress disorder [6,11]. These side effects may lead to reduced physical function and an inability to return to work [12,13]. Individuals with cancer also experience prolonged symptoms, for years and decades after cancer treatments have ended or their cancer has been treated into remission [14,15,16,17]. These symptoms are similar to post-sepsis syndrome and can include reduced joint range of motion, reduced strength, fatigue, reduced mobility, decreased physical function, anxiety, depression, poor quality of life, and frailty [16,17]. Those with a combined diagnosis of sepsis and cancer have increased risk of mortality and increased risk of living with substantial physical and psychosocial sequalae [1,18].

Rehabilitation interventions aim to optimize function and reduce disability and can include components of physical therapy, occupational therapy, speech language pathology, nutrition, and psychosocial therapies, which can be provided in acute hospital and outpatient settings [19]. Regular rehabilitation can minimize the physical and psychosocial effects of cancer [20,21,22,23,24] and sepsis [25,26,27], thereby improving the quality of life for survivors of both conditions. Despite this, studies examining the lived experience and satisfaction of survivors of sepsis or cancer regarding rehabilitation both during acute care and post hospital discharge have been conducted [28,29,30] but remain limited, highlighting the need for continued research. One study examined the suitability, extent, and satisfaction with rehabilitation in the year following a hospital admission for acute sepsis and found that patients believed that rehabilitation should begin earlier during their hospital stay, be more tailored to their specific physical outcomes, and include more detailed education [28].

While these studies provide preliminary evidence, more data are needed, especially regarding individuals with a history of both sepsis and cancer. Currently, there are no studies looking at the rehabilitation needs of individuals living with a previous diagnosis of both conditions. This information is important, because the severity of side effects and need for rehabilitation in those with sepsis and cancer is expected to be worse than the severity of one condition alone [31]. Further, the time when rehabilitation services are started (i.e., in the acute ICU setting versus in inpatients wards) is variable between geographical locations. Therefore, improving our knowledge on the bidirectional interaction of these conditions and the need for rehabilitation will help to improve the rehabilitation interventions provided to this population [18,31]. Therefore, the overall goal of this study was to examine the rehabilitation-related experiences of individuals with a history of sepsis and cancer and to explore the educational needs of sepsis survivors as well as of their caregivers and the healthcare team. To achieve this goal, we had two research objectives: (1) examine the rehabilitation-related experiences and barriers to rehabilitation for individuals with a history of sepsis and cancer, their caregivers, and the healthcare team; (2) explore patient, caregiver, and health professional needs for educational resources related to rehabilitation to improve the experiences of those with sepsis.

## 2. Materials and Methods

### 2.1. Study Design

We used a qualitative study design to achieve the research objectives. We performed a qualitative descriptive study [32,33] using interviews and focus groups. Qualitative descriptive studies are used when the goal is to provide a detailed description of a phenomenon or experience, and they prioritize the detailed description of an experience, rather than seeking to interpret it in depth [32,33]. They are particularly useful when little is known on a topic and provide insight for an initial understanding [32,33]. The Hamilton Integrated Research Ethics Board approved this study (ID: 15072), and we ensured written informed consent from all participants. We report the findings of this study in accordance with the COnsolidated criteria for REporting Qualitative research (COREQ) checklist for interviews and focus groups (see Appendix A) [34].

We undertook this study as part of the larger “ICaRe” project, which is an experience-based co-design (EBCD) project aiming to explore the Intersection of Cancer, sepsis, and frailty, and improve informational/educational tools describing the potential benefit of Rehabilitation services for these populations. EBCD is an approach involving multiple collaborators, including patients, caregivers, and health practitioners, sharing experiences of care at multiple stages during the design and implementation process to co-design products and services [35,36,37]. EBCD has been used in a wide range of clinical areas and is considered best practice for leading improvements in health services [37]. Figure 1 describes the EBCD process of the ICaRe project and the phases involved in this current study.

### 2.2. Participants

For the interview portion of this project, the study population included Canadians with a previous diagnosis of sepsis and cancer, their caregivers, or health professionals working with these populations. We included individuals with the following characteristics: (1) older than 18 years of age; (2) living in Canada; (3) able to understand English; (4) currently or previously diagnosed with sepsis as an adult and treated in hospital and previously diagnosed with cancer in adulthood (any type, stage, treatment status, or time since treatment for sepsis and cancer) or caregivers of someone with a current or past diagnosis of sepsis and cancer (any type, stage, treatment status, or time since treatment for sepsis and cancer) or healthcare professionals working with individuals with sepsis and cancer as part of their routine practice. For the interview portion of this study, sepsis did not have to occur during active cancer treatment, and the order of diagnosis of these conditions did not matter. For the focus groups, the study population included Canadians with a previous diagnosis of sepsis, their caregivers, or health professionals working with this population. We included individuals with the following characteristics: (1) older than 18 years of age, (2) living in Canada, (3) able to understand English, (4) currently or previously diagnosed with sepsis as an adult and treated in hospital or caregivers of someone with a current or past diagnosis of sepsis or healthcare professionals working with individuals with sepsis and cancer as part of their routine practice.

### 2.3. Recruitment

We used a mix of purposeful and snowball sampling to recruit participants for this project. Purposeful sampling included targeting community organizations supporting individuals with sepsis (i.e., Sepsis Canada and the Sepsis Alliance). These organizations received a study advertisement and email template to send out to their members, highlighting the inclusion criteria. Interested individuals were asked to email the study team to determine their eligibility and schedule an interview and/or a focus group. Further, we asked these same community organizations to post a recruitment advertisement for the interviews on their social media platforms. Finally, we purposefully sent an email to hospital staff who were known by the research team to work with patients with sepsis and cancer in Canada to see if they were willing to participate in this study. Once enrolled, we used snowball sampling by inviting the participants to suggest others to take part in the study.

### 2.4. Data Generation

We used two methods for collecting data: (1) individual virtual interviews; (2) virtual focus groups. Appendix A reports the interview and focus group guide. The use of multiple sources of data ensured data triangulation. We conducted the interviews first, from May to October 2023 to determine the rehabilitation-related experiences of individuals with a history of sepsis and cancer, their caregivers, and the healthcare team. These were one-on-one semi-structured interviews that lasted 30–60 min, where we asked the participants to discuss their experiences and needs related to living with sepsis and cancer, rehabilitation, or working with individuals with sepsis and cancer. The interviews were led by one of two study team members, including a critical care physiotherapist (A.N.L.N.), who was completing a post-doctoral fellowship at the time of the interviews, and a research assistant with previous experience in cancer rehabilitation research (J.P.), who was an MSc (physiotherapy) student at the time of the interviews.

Following the interviews, we conducted two separate focus groups where we shared an overview of the interview findings and explored the need for educational resources related to rehabilitation to improve the experiences of those with sepsis and cancer. Each interview was approximately 90 min long and occurred via Zoom in November and December 2023. During the focus group, we presented and discussed the interview findings and invited suggestions on the content and format of educational resources about these conditions and rehabilitation. The focus groups were led by a study team member (C.F.) who was a physiotherapist with clinical experience in the ICU and a PhD student. A research assistant (J.P.) took summary notes during the focus groups, provided verbal and written summaries of the discussion to the participants in the chat function on Zoom, and highlighted the overall atmosphere and feel of the focus groups for the study team. The use of multiple researchers to collect and analyze data ensured investigator triangulation within this study.

### 2.5. Sample Size

We aimed to recruit 10–20 interview participants and 12–16 participants for the focus groups [35]. In order to achieve adequate recruitment to answer our research questions, we ensured saturation of data from the entire sample as a whole, confirming no new categories were emerging from the data, prior to stopping recruitment [38]. No criteria limited the interview participants from taking part in the focus groups.

### 2.6. Data Analysis

We recorded interviews and focus group discussions and transcribed the recordings verbatim, initially by using the Zoom transcription function. Two reviewers (J.S-T and J.P.) analyzed the interview and focus group transcripts using qualitative content analysis and inductive coding [39]. We used independent line-by-line analysis to generate initial codes and consensus to finalize and develop the definitions for codes. The codes were then grouped and consolidated into categories. Independently, the investigators reviewed each other’s coding of the transcripts to assure credibility. We extracted quotes from the transcripts that corresponded to the established codes and categories. To organize and manage the qualitative data, we used NVivo (Version 14).

## 3. Results

### 3.1. Participant Description

We included 17 individuals for the one-on-one virtual interviews. One participant withdrew from the study following the interview due to concerns about how their information would be received by their treating healthcare team, and another was removed following their interview due to our team’s concerns regarding their eligibility (i.e., during the interview, they were not able to answer simple questions related to their sepsis and cancer diagnosis, and conflicting information was given in their responses regarding this). Therefore, we included 15 participants in the analysis (three patients with both sepsis and cancer, six caregivers of patients with sepsis and cancer, six healthcare professionals working with individuals with sepsis and cancer). Most participants (*n* = 12; 80%) were female, and the mean participant age was 43 years. Of the patient participants, one was previously diagnosed with Hodgkin’s lymphoma, one with acute myeloid leukemia, and one with bladder cancer, and all were treated for sepsis as inpatients. The mean length of time since sepsis diagnosis was 6.4 years. The mean length of time since cancer diagnosis was 14 years. For the included caregiver participants, two cared for individuals with prostate cancer, and one each for individuals with non-Hodgkin’s lymphoma, chronic lymphocytic leukemia, lung cancer, and breast cancer. The mean length of time since sepsis diagnosis for the individual they were caring for was 2.5 years. The mean length of time since cancer diagnosis for the individual they were caring for was 5.5 years. The health professionals included two nurses, one medical oncology fellow, one physiotherapist, one occupational therapist, and one medical doctor. All health professionals worked in inpatient settings and were currently practicing.

Fifteen participants took part in the focus groups (six patients—one with sepsis and cancer, five with sepsis only, all treated for sepsis as inpatients—three caregivers, and six healthcare professionals who worked with inpatient populations); the majority were female (*n* = 14, 93%). One of the patient participants disclosed a previous cancer diagnosis (Hodgkin’s lymphoma, diagnosed 22 years ago). The mean length of time since sepsis diagnosis for the focus group patient participants was 6.3 years. Two caregivers cared for an individual with sepsis and cancer (prostate cancer and non-Hodgkin’s lymphoma). The mean length of time since sepsis diagnosis for the individual they were caring for was 2.3 years. The mean length of time since cancer diagnosis for the individual they were caring for was 3.5 years. The health professionals included two physiotherapists, two registered dietitians, one occupational therapist, and one nurse practitioner. Table 1 provides the characteristics of the included individuals by participant type.

### 3.2. Experiences with and Barriers to Rehabilitation (Research Objective 1)

We identified three main themes related to rehabilitation for individuals with sepsis and cancer: (1) the rehabilitation experience did not meet the patient’s care expectations; (2) barriers to rehabilitation exist on multiple levels; (3) there are important gaps in education on how to improve physical and psychosocial outcomes.

#### 3.2.1. The Rehabilitation Experience Did Not Meet the Patient’s Care Expectations

The participants highlighted that the rehabilitation they received during their care for sepsis and cancer ‘missed the mark’. Both patient and caregiver participants felt that once individuals were discharged from the hospital, they were on their own to find and pay for rehabilitation services.

“So, it was basically…one long period of time [in hospital] of physio, OT [occupational therapy], SLP [speech language pathology], dietitian, social work, and then once I was home, obviously those things…were very intermittent in the services I was receiving”.(Patient, female)

“We had to pay for an occupational therapist to come see my mom when she was discharged to…talk about whatever it is occupational therapists do! And then…she had physio, I suppose, come to the house every week or so to give her exercises to do”.(Caregiver, female)

Many health professionals mentioned that rehabilitation was valued by the healthcare team but was provided inconsistently across the phases of care.

“If I have a patient in front of me who I deem needs admission because of their current medical or frailty status…then yes, the inpatient admitting service would make arrangements for the appropriate allied health services [e.g., physiotherapy (PT), OT] to see those patients…my main concern is the patients who go home and what they’re able to access. It’s a comfort sometimes when I can admit a patient and know that they’ll be seen by my inpatient [team]”.(Health professional, female)

“[The] ICU doesn’t utilize OT because it’s so early in the course of treatment that OT isn’t necessarily the most appropriate treatment…I see people in their sickest, most acute phases, so I don’t get to see much that happens from the rehab point of view once they leave the intensive care setting”.(Health professional, female)

It was highlighted by all participant groups that a multi-disciplinary approach to care, including rehabilitation professionals, is needed to improve the outcomes of those with cancer and sepsis.

“We always order PT and OT because they need to get going. It’s part of our admission package, which is great...I can’t think of an example where I didn’t use PT, OT, actually”.(Health professional, male)

“Sepsis is head-to-toe systemic, so I think that for the physiotherapists and occupational therapists to really have an opportunity to study and research ways that would improve the quality of lives for patients…and…nutrition…the right combination of strengthening and conditioning [is needed and important]”.(Caregiver, female)

#### 3.2.2. Barriers to Rehabilitation Exist on Multiple Levels

The participants described a wide variety of barriers to rehabilitation during and after care for cancer and sepsis. They discussed external and internal influences that affected the availability of rehabilitation services. External influences included COVID-19 and the availability and costs of community rehabilitation services. These ideas were brought forth in statements such as:

“My mom had a lot of barriers because she had cancer and sepsis during COVID, so I think there was a lot less opportunities to go see someone. It had to be in the hospital or at the house. My mom wasn’t allowed to leave the house because of the potential of getting COVID during cancer…so my mom just didn’t leave the house. So, it was tough for my mom to get that if it wasn’t offered at the hospital specifically”.(Caregiver, female)

“We only had an OT maybe once in the hospital, and then we paid for someone at home to come visit her, but it wasn’t much…we didn’t see them as often…it would be more helpful if they could come every day—I mean, and I get it funding and money [make it difficult]”.(Caregiver, female)

“When you transition from peds [pediatrics] to adult, it’s like they throw you to the wolves, and if you don’t know how to advocate for yourself, you’re kind of [in trouble] because advocating for appointments, advocating for services that you might need, advocating for a diagnosis or a test or whatever…if you don’t know how to advocate for yourself when you’re an adult, you’re kind of left to fend for yourself”.(Health professional, female)

Internal influences included health professional time in hospital, healthcare professional burnout, and workplace culture. These influences were evident in statements such as:

“They were too busy, number one, to spend much time [with us]. They had such a huge roster, and they were not all equally educated in the treatment and management of potential sepsis”.(Caregiver, female)

“COVID made it worse because of the burnout, because of staffing…the acuity of patients we had with COVID…was very labor-intensive. The additional mental, emotional, and physical demands that COVID brought added to fatigue and burnout levels, which affected staffing”.(Health professional, female)

“There are challenges in some units, our units included, where there may be…a culture that mobility is more of a—it’s not as necessary as we believe it is, so we often, as physios, have to really advocate for our patients that mobility is really an important piece to prevent them from declining and to allow them to have the best quality of life once they’ve recovered from their critical illness”.(Health professional, male)

The impact of social support affecting the ability to participate in rehabilitation was another internal barrier highlighted in statements such as:

“[The nurses]… they have other patients, and so my mom probably wasn’t getting the encouragement [to do her exercises], or she just wanted to sleep. When she was at home, my dad could be like, ‘okay, we’re going to do X’ or me, ‘Okay, mom, let’s do some exercises”. She had that encouragement and someone to also spot her if she was to get weak or fall”.(Caregiver, female)

“The biggest challenge…is that they don’t necessarily get the intensive support that we provide in the ICU setting once they hit the wards, so I think there’s a bit of a disconnect there and some disappointment that they’re not getting the same level of physiotherapy or rehab once they leave our unit”.(Health professional, female)

#### 3.2.3. There Are Important Gaps in Education on How to Improve Physical and Psychosocial Outcomes

The participants described gaps in the education they received that were beyond the field of rehabilitation, which affected them physically and psychologically during and after treatment for sepsis and cancer. One common theme was that sepsis education was delayed, which led to fear and anxiety among patients and caregivers. It was also brought forth repeatedly that patients and caregivers had to independently search for information on sepsis and/or the link to cancer. This was highlighted in statements such as “*the actual diagnosis of sepsis, I learned all of it on my own*” (caregiver, female) and in others such as:

“I had just looked up sepsis, and I came across the foundation. And then that helped me a little bit too…but…you start searching everywhere and everything to try to understand every single aspect”.(Patient, female)

“When my mom was going through sepsis or septic shock, they were trying so hard to save her life, we were only given access to the nurses… they can’t tell me why this happened or the reason why my mom got sepsis, or the connection that it had to cancer…the information that I’ve learned from sepsis and septic shock is from…Sepsis Canada and learning about everything after. To be honest, when my mom was diagnosed with sepsis, I had no idea what that is or what it meant. My dad had no idea. My brothers had no idea. So, I think in general, sepsis is just not very well known or talked about even though it’s insanely serious and common”.(Caregiver, female)

Health professionals said that they had a lack of entry-level education on these conditions and the role of rehabilitation in improving outcomes, which affected the care they provided: “*In my original entry-level physio training we covered… I think we might have had one lecture on cancer*” (health professional, male). Further:

“During my formal education…as a medical student, I would say minimal to none [information on rehab for cancer and/or sepsis]. Because I did do electives in physiatry and geriatrics, I had a bit more exposure, but those were not mandatory rotations by any means. So, you know, some of my colleagues who went through with me would have had even less formal training”.(Health professional, female)

### 3.3. Patient and Caregiver Education Needs (Research Objective 2)

We identified two main themes related to focus group participants’ needs for educational resources to improve the experiences of those with sepsis: (1) educational content provided must be specific and meaningful; (2) multi-modal resources are needed to suit diverse stakeholder needs.

#### 3.3.1. Educational Content Provided Must Be Specific and Meaningful

The focus group participants identified many ideas describing the content that should be included in future educational resources (see Table 2 for a list of suggested content). These participants highlighted that content must be short and easy to digest, include examples, and normalize the experience of having to regain function post sepsis, as follows:

“Simple stuff that’s going to stick in your head that doesn’t take a ton of time to process/to think about…that you can stick up on your wall, that you can have posted in a cancer center. Something like that could be helpful”.(Patient, female)

“I would personally love to see examples. You know, I was very frustrated. I wasn’t given rehab in the hospital and when I came home…I had worked out a lot before and all of a sudden…I would faint every time I walked…really specific examples of what can be done [are needed]. You know, from bed. What can be done from seated?”.(Patient, female)

“I think it’s an important thing to let people know what they’re experiencing, the frustrations [with function] are okay…just keep at it and don’t rush and you can do it”.(Patient, female)

“[We need to] educate [survivors or sepsis and/or cancer patients] on the possible outcomes so that …they know what to look for and they know what’s coming”.(Health professional, female)

The focus group participants also described the need for different educational resources for different partner types (i.e., different information for healthcare professionals, patients, and caregivers).

“The language in healthcare and the language in the community are two different languages”.(Patient, female)

“Medical terminology for sure for those medically educated…And then of course breaking down in layman’s terms, very simple [for patients]… because when you’re sick, you can’t really decipher everything…so make it really simple for those that are going through it”.(Patient, female)

#### 3.3.2. Multi-Modal Resources Are Needed to Suit Diverse Partner Needs

The focus group participants highlighted the need for a wide variety of modes of education to suit individual patient, caregiver, and health professional needs (Table 2):

“I think the reality is a lot of people learn in different ways and a lot of people [have unique needs]… at certain phases you need something simple like that beautiful brochure that’s simple and easy to read and then you want more content so the education modules might be helpful…you know where you’ve been and where you’re going”.(Patient, female)

“Educational material should be available online but also I think that it’s something that can be handed out in the hospital for family to be able to read up on. I really wished that I was more informed about the reality of sepsis. It wasn’t until I was home and googled sepsis that I truly understood the severity of it”.(Patient, male)

“Written is helpful because it allows people to refer back to it, but truly, because everybody is so unique, I think there has to be someone that’s able to sit down and explain things to people…tailor it to their unique situation a little bit”.(Health professional, female)

## 4. Discussion

This qualitative study provides preliminary evidence addressing the specific needs of individuals with cancer and sepsis, their caregivers, and the healthcare team related to rehabilitation during and after treatment. The results suggest that the rehabilitation services received by our participants did not meet their needs, and that this population, their caregivers, and the healthcare professionals had barriers to rehabilitative care including the systemic nature of sepsis, the COVID-19 pandemic, access to rehabilitation services, lack of social support, and workplace culture. This study also explored the education needs of individuals with sepsis and their caregivers. The results demonstrate the need for multi-modal education resources highlighting content in clear and concise ways to inform individuals with sepsis and cancer on the link between these conditions and the rehabilitation strategies to improve outcomes.

Consistent with previous studies in cancer rehabilitation [40,41,42,43], access to services was as a major barrier to rehabilitation during and after treatment for sepsis and cancer. Specifically, after sepsis, the participants in this study described having to seek out and independently pay for rehabilitation services. This is consistent with a 2023 study [28] that surveyed survivors of sepsis recently discharged from the ICU and their caregivers on unmet needs and found that access to timely rehabilitation during and after hospital admission was a primary unmet need of the respondents. However, in our study, additional barriers to rehabilitation were also identified, including external influences (i.e., COVID-19) and internal barriers, such as healthcare professional burnout and workplace culture. These are likely due to the timing of this study and the lingering effects of the pandemic on the healthcare system in Canada, including staff shortages, changes to public health measures, and increased wait lists for procedures [44,45,46]. During the pandemic, many healthcare professionals were required to take on roles outside of their normal duties, and therefore, decreased access to quality care has been described by patients [47,48]. While the pandemic may be over, research in Canada demonstrates that burnout among public healthcare workers continues to be high (almost 79% of the healthcare workers surveyed said they were burnt out; 88% suffered from disengagement) [49]. Overcoming these continued obstacles for rehabilitation professionals requires focus to ensure access to equitable care [49]. It was also evident from this study that many patient and caregiver participants were not aware of the need for rehabilitation during or after a sepsis diagnosis. This lack of knowledge on the benefit of rehabilitation to manage the impacts of a medical diagnosis is common in many other conditions [41,50], and therefore the need to create mechanisms to increase knowledge on how to improve outcomes is an area for future research. Additionally, information not only on the benefit of rehabilitation services but also on ‘how to’ implement rehabilitation into daily life, including exercise intensity, timing, parameters, and progressions, was suggested as necessary in this study and is an area of needed consideration by clinicians.

While the aim of our study was to explore the educational needs related to rehabilitation of those with a previous diagnosis of sepsis, a lack of general knowledge on what sepsis is, why it happens, the link between sepsis and cancer, and post-sepsis outcomes, such as frailty, came forth consistently. This lack of or delayed knowledge was described as leading to fear and increased anxiety in those with these conditions and their caregivers. Early education, using any media, can reduce patient anxiety [51,52]. Further, many of our participants described needing to search for information independently and learn about complications and the sequelae on their own and questioned whether the information that they found was accurate. Our study supports the need for more education on sepsis and its link to cancer to improve patient experience and outcomes.

From the healthcare professional standpoint, many shared that they received little information regarding sepsis, post-sepsis syndrome, or rehabilitation during their training. While it is unclear why this gap exists, it is clear that a lack of knowledge on sepsis and the benefit of rehabilitation post sepsis diagnosis can prevent or cause delay in appropriate treatment and lead to the developed of significant side effects limiting function [25,26,28]. As with past research [50,53], this highlights the need to also create educational resources for healthcare professionals regarding the intersection of these conditions and the potential for rehabilitation to improve outcomes. Our study identified a knowledge gap for healthcare professional on the available evidence-based community rehabilitation resources for patients, consistent with previous literature for individuals with cancer [50,54,55]. Therefore, in addition to assessment and basic management techniques, an inventory of available resources and easy referral pathways to help patients access rehabilitation care is needed.

This project is part of a larger EBCD process [35,37], looking to create meaningful high-quality educational resources for individuals with a diagnosis of sepsis and cancer. An important foundational component of this phase of research engages diverse partner perspectives and considerations for creating resources in follow-up phases. Consistent with other literature examining patient education resources in healthcare [51,52], these recommendations include multi-modal rehabilitation resources (i.e., paper and electronic) of different lengths for individuals at different points of care. Further, general reminders and prompts on what sepsis is and how to identify it were recommended by our focus group participants. Healthcare prompts are tools that help to deliver messages to motivate individuals to change behaviors [56,57]. They have previously been used successfully in the cancer rehabilitation literature to increase movement and physical activity discussion [58,59]; however, there are no studies examining the use of prompts in relation to sepsis, highlighting an area of future research. Overall, the participants highlighted that the content needs to be ‘readable’, clear, and easy to digest.

### Strengths and Limitations

The strengths of this study include the novelty of the research question, the inclusion of diverse participants, and the use of data triangulation (through both interview and focus group methodology) and investigator triangulation (different researchers collected and analyzed the data), which together provided meaningful results to build on in future work. Further, the iterative content analysis using multiple reviewers strengthens the validity of the methods used in this qualitative study. However, the findings should be viewed with an understanding of their limitations. All participants resided in Canada, and most of the patient participants were diagnosed and treated in Ontario. Most individuals with a sepsis diagnosis in this study described their experience with rehabilitation as inpatients and reported a lack of outpatient rehabilitation services. Therefore, further research needs to explore outpatient rehabilitation strategies post hospital discharge. Further, this study is limited by its small sample size and the fact that most of our participants identified as female, which may be due to our recruitment procedure and methods used (interviews and focus groups), which have been shown in the literature to be more acceptable by female participants and may have led to sampling and sex/gender biases. Additionally, our recruitment methods may not have reached potentially eligible people who do not access social media or are not on the email lists of our partners. We also collected data using virtual means only, which may have limited the participation of those without access to technology and the study generalizability. Additionally, the recruitment of ineligible participants using social media has become a problem in data generation [60,61]. Our study team experienced many ineligible participation requests in this study and had to diligently screen to ensure eligibility. Additionally, while we ensured that saturation of data occurred within our entire sample during analysis, we were not able to ensure saturation by individual participant type or characteristics (i.e., cancer type, time since the diagnosis of sepsis or cancer, health professional type), as our participants included health professional types and individuals with different stages of cancer that were only represented by one participant. Further, we did not collect social or cultural demographic data and therefore could not analyze our data based on these criteria. Therefore, this study did not describe specific problems related to those with specific types of cancer or sub-types of sepsis, but only addressed general patient issues, which future research can build upon in larger populations. Hence, these results must be interpreted with this consideration. Finally, while this study focused on sepsis and cancer, it became clear that rehabilitation post sepsis in general was of utmost interest to many of the participants and is an area that is relatively unexplored. Future research should focus on developing rehabilitation strategies for this specific population.

## 5. Conclusions

We identified gaps in the care of the participants in our study. Rehabilitation was identified as an area of future need by all participants. The interview participants in this study were unaware of the benefit of rehabilitation during and after sepsis and cancer and did not know what to do to regain function and improve overall outcomes. We identified important barriers to rehabilitation for this population related to lack of knowledge, lack of available resources, healthcare settings, the pandemic, and workplace culture. This study highlights the need to create educational resources for diverse participants focusing on rehabilitation for those with sepsis and cancer to improve outcomes and highlighting the importance of the link between these two conditions. These resources should be specific and provide information on how to access and engage in rehabilitation at various points across the care trajectory. Future research should build on this by engaging a larger representation of patients with sepsis and cancer.

## Figures and Tables

**Figure 1 healthcare-13-00822-f001:**
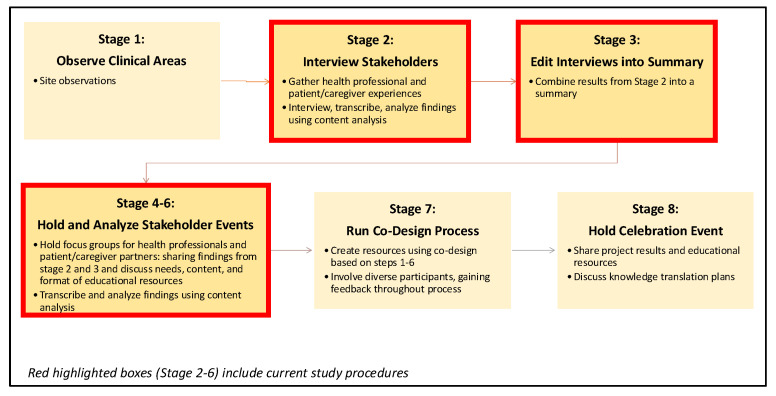
Experience-based co-design process of the ICaRe project.

**Table 1 healthcare-13-00822-t001:** Characteristics by participant type (total N = 30).

Characteristic	Patient (*n* = 9; 4 Diagnosed with Sepsis and Cancer, 5 with Sepsis Only) *n* (%)	Caregiver (*n* = 9) *n* (%)	Healthcare Professional (*n* = 12) *n* (%)
Sex			
Male	2 (22%)	0	2 (17%)
Female	7 (78%)	9 (100%)	10 (83%)
Age (Mean, SD) yrs	45.13, 10.27	44.75, 14.94	41.33, 7.63
Cancer Type *			
Hodgkin’s Lymphoma	2 (22%)	-	
Non-Hodgkin’s Lymphoma	-	2 (22%)	
Acute Myeloid Leukemia	1 (11%)	-	
Chronic Lymphocytic Leukemia	-	1 (11%)	
Bladder Cancer	1 (11%)	-	
Breast Cancer	-	1 (11%)	
Prostate Cancer	-	3 (33%)	
Lung Cancer	-	1 (11%)	
Time since cancer diagnosis * (mean, yrs)	16	5 (relates to individual they were caring for)	
Time since sepsis diagnosis * (mean, yrs)	7	2 (relates to individual they were caring for)	

* Note: questions answered for self or in relation to care recipient.

**Table 2 healthcare-13-00822-t002:** Suggested content for future educational material.

Education Content for Patients and Caregivers	Education Content for Health Professionals	Mode of Education
Available support groups for individuals post sepsis	Awareness of personal biases and how they influence care	E-modules for continuous review
Common signs and symptoms during and after sepsis	Communicating with patients with sepsis and cancer	Social media posts
How to recognize sepsis	Discharge considerations	Visuals: handout/poster/brochure; information card; wall stickers; postcard; tip sheet for family members
How to talk to your family about sepsis	Evaluation tools/outcome measures to use with this population	
How to talk to health professionals about cancer and sepsis	Information on post-sepsis syndrome	
Link between sepsis and frailty	Sleep education	Short videos: commercial
Link between sepsis and cancer	Sepsis general information (What is it? How to recognize it?)	Website housing all material to make it easy to access
Mental wellbeing during and after sepsis	Pain management strategies	
Nutritional advice	Screening for rehabilitation and diet needs	
Preventing sepsis in those with cancer (i.e., diet, exercise, other lifestyle modifications)		
Prognosis (physical and health-related) post sepsis		
Rehabilitation: benefits during and after cancer and sepsis		
Rehabilitation: what exercises to do and how to monitor intensity of exercise		
Rehabilitation: managing expectations		
Where to ask questions/get more information post cancer and sepsis		

## Data Availability

The data supporting the findings of this review are available from the corresponding author, J.S.-T., upon reasonable request.

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
