# Peer review of "A Qualitative Study Exploring the Rehabilitation Experience of Individuals with a Previous Diagnosis of Cancer and/or Sepsis, Their Caregivers, and Health Providers"

_healthcare, 2025, doi:10.3390/healthcare13070822_

Round 1

Reviewer 1 Report

Comments and Suggestions for Authors

Dear Authors,

Thank you for providing me with the opportunity to read this interesting paper. Below, I have listed my comments:

1) While the introduction explains the impact of sepsis and cancer, it does not clearly establish the knowledge gap this study aims to fill. what is missing in current research? what is the novelty of your study?

2) The data analysis section does not need to be in italics.

3) In the discussion, a lack of knowledge among healthcare professionals is identified but it is not clear why this gap exists or how it impacts patient outcomes.

4) Do the issues mentioned related to COVID-19, persist in the post-pandemic era or have they been improved? It is worth exploring that giving we have passed the covid era.

I hope this feedback is helpful.

Reviewer 2 Report

Comments and Suggestions for Authors

The objectives should not appear in the Methodology section.
The study design section contains information that corresponds to other sections of the Methodology but does not define the study design, for example, having followed the COREQ checklist, or the flow chart in Figure 1. But the design is not clearly defined.
The table with the primary and secondary study categories and their relationship with the questions that guided the interview and the focus groups should be attached. 
The study population is not defined
The snowball sampling technique is applied when the study population is difficult to access, not very visible or not well defined, and when the participants can refer to other members with similar characteristics, and for example when researching phenomena of a certain invisibility such as migration or family and gender-based violence. In these cases, recruitment is done through informal social networks. However, this sampling technique is not well suited to this study, nor is it justified insofar as Sepsis Canada and the Sepsis Alliance are institutions that support the research and it is through them that the individuals under study are accessed. The type of sampling is obviously different. This needs to be corrected. 
An important limitation is that the collection of information was not done face-to-face in any case, which, in the focus group, may be a limiting factor. At least some of the interviews and the focus groups should have been conducted in person.

There is no indication of the time or dates when the study was carried out. If the fieldwork was carried out after the pandemic, there is no justification for not collecting face-to-face data at any time.Was theoretical triangulation or triangulation by different researchers carried out? It appears in the limitations very superficially, but is not expressed in the methodology. 
If all individuals are women, there is a gender bias.The criterion of homogeneity has not been adequately handled.At the very least, the perspective of the gender variable should be included in the analysis of the data, since this is not a general population, only women. 

Social and cultural factors should have been included in the data analysis. Age, role (patient, caregiver, family member..) and type of cancer are not enough to analyse the phenomenon deeply under a social and cultural perspective and its influence in health

Round 2

Reviewer 2 Report

Comments and Suggestions for Authors

-Previous comment (nº 2): “ The study design section contains information that corresponds to other sections of the Methodology but does not define the study design, for example, having followed the COREQ checklist, or the flow chart in Figure 1. But the design is not clearly defined”.

Part of your answer has been:  “We moved the information regarding EBCD and where this study fits into our larger project to the introduction (see paragraph starting line 94 – moved up from previous). In past publications we have listed the reporting guidelines as part of the study design section. They don’t appear to fit anywhere else under the headings suggested by this journal, but I’m happy to move them if you suggest where reference to the COREQ guidelines would be more appropriate.”

I didn’t suggest to move to the Introduction section anything about the stages of the research, nor the allocation of this study in the whole project. I just said that the definition of the study design can’t be done with the stages of the research nor the application of a checklist. I am talking about one thing and you have answered something completely different, moving information that should still be in Methodology, but which in no way defines the design of the study. Related to this topic, the tools to obtain the information don’t define the study design. The ethics approval doesn’t define the study design. The use of COREQ doesn’t define the study design.

- Previous comment nº 4: You talk about the sample, sampling procedures and inclusion/exclusion criteria, but you do NOT define the study population. You continue without defining the study population.

Author Response

Thank you for the updated comments. Please see our responses below.

COMMENT 1: Previous comment (nº 2): “ The study design section contains information that corresponds to other sections of the Methodology but does not define the study design, for example, having followed the COREQ checklist, or the flow chart in Figure 1. But the design is not clearly defined”. Part of your answer has been:  “We moved the information regarding EBCD and where this study fits into our larger project to the introduction (see paragraph starting line 94 – moved up from previous). In past publications we have listed the reporting guidelines as part of the study design section. They don’t appear to fit anywhere else under the headings suggested by this journal, but I’m happy to move them if you suggest where reference to the COREQ guidelines would be more appropriate.” I didn’t suggest to move to the Introduction section anything about the stages of the research, nor the allocation of this study in the whole project. I just said that the definition of the study design can’t be done with the stages of the research nor the application of a checklist. I am talking about one thing and you have answered something completely different, moving information that should still be in Methodology, but which in no way defines the design of the study. Related to this topic, the tools to obtain the information don’t define the study design. The ethics approval doesn’t define the study design. The use of COREQ doesn’t define the study design.

RESPONSE: Thank you for the clarification. We have returned the information on EBCD to the methods section. We have added a sentence to the study design section highlighting that this is a qualitative research study and left the information in this section on the type of qualitative design used (i.e., a qualitative descriptive study) which was there previously and referenced. Please see starting line 95 of updated version. "We used a qualitative study design to achieve the research objectives. We performed a qualitative descriptive study [32,33] using interviews and focus groups. Qualitative descriptive studies are used when the goal is to provide a detailed description of a phenomenon or experience, and prioritizes the detailed description of an experience, rather than seeking to interpret it in depth [32,33]. It is particularly useful when little is known on a topic and provides insight into initial understanding [32,33]." If we are still misunderstanding this question, please provide more information on what you suggest we add here.   

COMMENT 2: Previous comment nº 4: You talk about the sample, sampling procedures and inclusion/exclusion criteria, but you do NOT define the study population. You continue without defining the study population.

RESPONSE: We have added in two sentences in the participants section that highlight the study population. This is followed by the eligibility criteria. See highlighted lines starting 118. Again, we want to adequately address your concern, so if we are still misunderstanding, please provide more information on what you suggest we add here. 

Thank you for this review. 

Round 3

Reviewer 2 Report

Comments and Suggestions for Authors

.